# IOMM: Fast Pre-training of Unified Multi-modal Models without Text-Image Pairs

## Abstract

Unified Multimodal Models (UMMs), which integrate deep visual understanding with generative capabilities, are often constrained by inefficient training paradigms and a heavy reliance on scarce, high-quality text-image paired data. In this paper, we systematically analyze existing pre-training recipes for UMMs and identify these two issues as major bottlenecks. To address them, we propose **Image-Only Training for UMMs (IOMM)**, a data-efficient two-stage training framework. The first stage pre-trains the visual generative component using abundant unlabeled image-only data, thereby removing the dependency on paired data. The second stage fine-tunes the model using a mixture of unlabeled images and a small curated set of text-image pairs, leading to improved instruction alignment and generative quality. Extensive experiments show that IOMM not only improves training efficiency but also achieves state-of-the-art performance. For example, our base model IOMM-B, trained generation module from scratch purely on open-source data using approximately only **1050** H800 GPU hours (including **1000** hours for image-only pre-training), attains a score of **0.89** on the GenEval benchmark—surpassing strong baselines such as BAGEL (0.88) and BLIP3-o (0.84). Code will be released publicly.

## 1 Introduction

Unifying deep semantic understanding with rich perceptual generation in a single model is a grand challenge in AI. These Unified Multimodal Models (UMMs) promise a synergy where comprehension and generation mutually enhance one another, unlocking applications from nuanced, dialogue-based image editing to context-aware content creation (Google, 2025a;b; OpenAI, 2025). While recent UMMs demonstrate impressive generative capabilities (Wu et al., 2025a; Chen et al., 2025a; Pan et al., 2025; Dong et al., 2024), their development is often hampered by significant practical constraints.

However, current UMM training paradigms rely on vast, often proprietary, text-image datasets (Chen et al., 2025a). The prohibitive cost of curating this data impedes open and reproducible research. Moreover, the training procedures are notoriously inefficient, demanding immense computational resources. This raises a critical question: *Can we develop a more data- and compute-efficient training paradigm for UMMs that reduces reliance on paired data while improving performance?*

In this work, we address this question by deconstructing the pre-training of UMMs' visual generative components. Our analysis reveals two primary bottlenecks: the dependency on scarce text-image pairs and the inefficiency of prevailing training objectives. We observe that many UMMs, particularly when fine-tuned on limited data, struggle to generate images that faithfully align with textual prompts. As shown in Fig. 1a, even a strong baseline like Qwen-Image (Wu et al., 2025a) can produce outputs that lack detail and fidelity to the input prompt.

To surmount these limitations, we introduce IOMM, a novel, data-efficient two-stage training paradigm for constructing and refining UMMs. Our approach commences with an unsupervised pre-training phase that leverages unlabeled, image-only data, followed by a fine-tuning stage that employs a strategic mixture of image-only and high-quality paired data. This paradigm, as we empirically demonstrate, not only mitigates the reliance on paired data but also yields superior generative quality and instruction-following capabilities. **In summary, our contributions are threefold:**

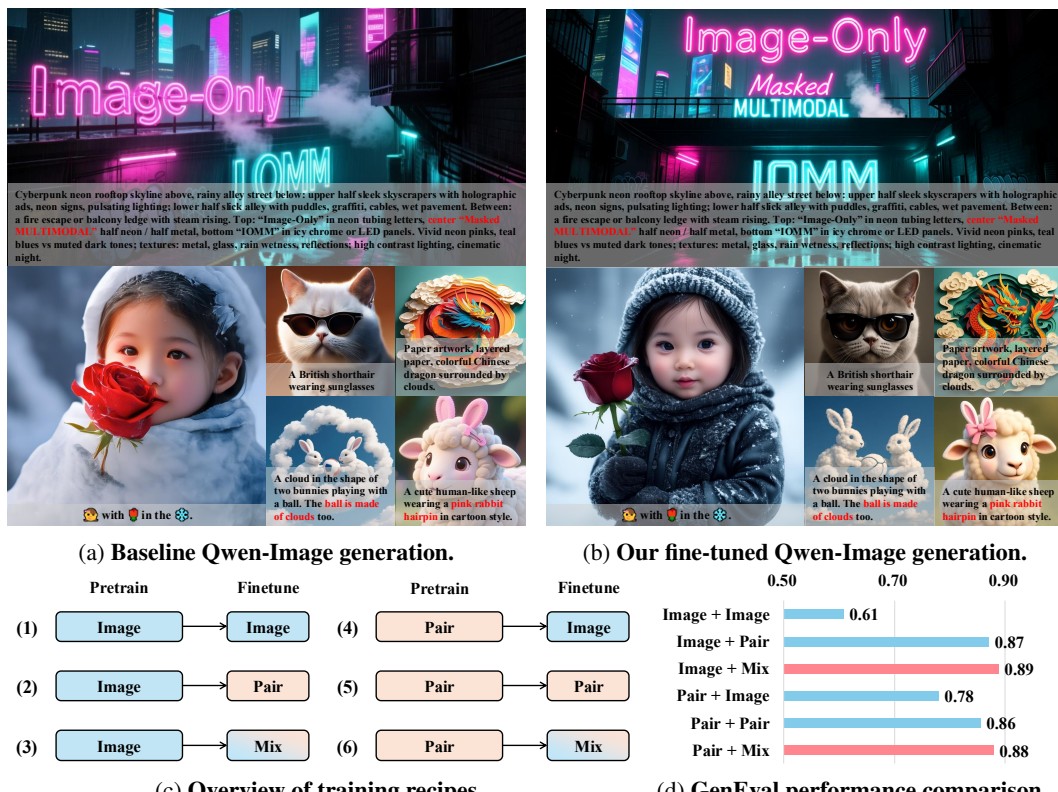

(a) **Baseline Qwen-Image generation.**

(b) **Our fine-tuned Qwen-Image generation.**

(c) **Overview of training recipes.**

(d) **GenEval performance comparison.**

Figure 1: **An overview and validation of our proposed training paradigm. (a, b)** Qualitative comparison between the original Qwen-Image model and our fine-tuned version. Our method enhances the model's ability to generate images with richer visual detail and improved alignment to the textual prompt. **(c)** An illustration of the six training recipes we investigate. **(d)** Quantitative results of six training recipes on the GenEval benchmark.

(a) We present a systematic analysis of six distinct training recipes for UMMs, exploring various combinations of image-only, text-image pair, and mixed data across pre-training and fine-tuning. Our central finding is that a two-stage paradigm—pre-training on image-only data followed by fine-tuning on a mixed dataset—yields best performance (Fig. 1d).

(b) We introduce IOMM, a data- and compute-efficient framework built upon two key technical innovations: (1) a novel *residual query adapter* that efficiently adapts frozen Multimodal Large Language Models (MLLMs) for generative tasks with minimal parameter overhead, and (2) a *masked image modeling* objective that fosters a robust visual prior by framing pre-training as a sparse-to-dense reconstruction task.

(c) Extensive experiments validate the efficacy and efficiency of IOMM. Our resulting models attain SOTA or comparable performance across diverse benchmarks, architectures, and resolutions, all while operating with substantially greater data and compute efficiency (see Sec. 4).

Additionally, we establish that our proposed *mixed-data fine-tuning strategy* is a generalizable and effective technique for enhancing the instruction-following fidelity and image generation quality of existing powerful UMMs, which we validate on diverse models including Qwen-Image (Sec. 4.3).

## 2 RELATED WORK

**Text-to-image diffusion models.** The field of text-to-image synthesis has seen rapid advancements, driven by innovations in diffusion model architectures and training methodologies. Foundational works, such as the initial Stable Diffusion series (Rombach et al., 2022; Podell et al., 2024), established the Latent Diffusion Model (LDM) as a dominant paradigm. A significant architectural evolution arrived with Stable Diffusion 3 (Esser et al., 2024), which introduced the Multimodal Diffusion Transformer (MM-DiT). This architecture employs separate transformer-based pathways to

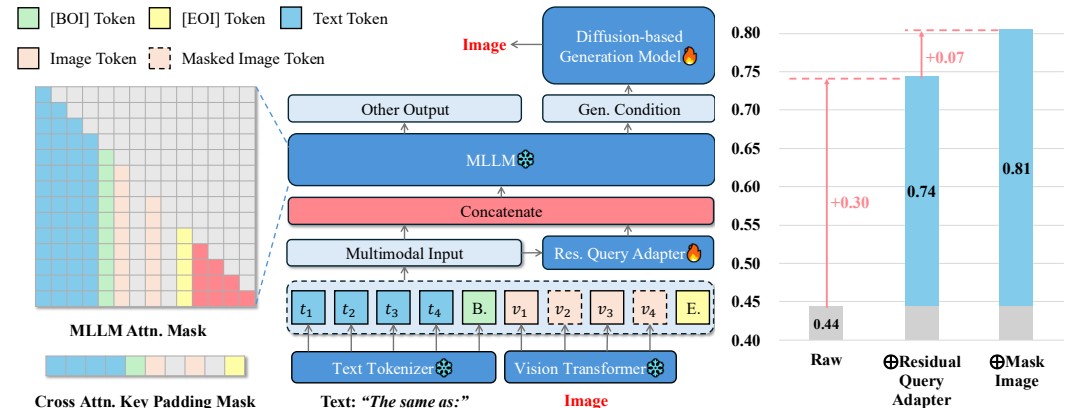

(a) **The architecture of our image-only pretraining stage.**  (b) **Component ablation study.**

Figure 2: **Visualization of the IOMM framework. (a)** The architecture of our proposed framework. **(b)** An ablation study demonstrating the effectiveness of our architectural design choices, confirming that each component contributes positively to the final performance.

process image and text representations independently before fusing them, markedly improving text-image alignment. Following a similar design philosophy, FLUX.1 (Labs et al., 2025b) also utilizes a dual-stream transformer architecture to enhance modality-specific encoding.

Concurrently, a parallel line of research has focused on optimizing training efficiency and data curation. For example, PixArt-$\alpha/\sigma$ (Chen et al., 2024b;a) demonstrated the ability to achieve state-of-the-art performance with substantially reduced training costs. Similarly, Playground v2/v2.5 (Li et al.; 2024) is distinguished by its high aesthetic quality, a result of meticulous data filtering and reinforcement learning from user preferences. More recent models, including SANA (Xie et al., 2024) and SANA-sprint (Chen et al., 2025b), continue this trajectory, pushing the boundaries of performance through further architectural and training refinements.

However, these models are specialized for unidirectional text-to-image generation. They lack the inherent capacity for multimodal understanding, which precludes their direct application to complex, interactive tasks such as dialogue-based image editing (Wu et al., 2025a; Google, 2025a) that require a seamless blend of comprehension and generation.

**Unified understanding and generation models.** The pursuit of models that unify multimodal understanding and generation has led to two primary training paradigms: training end-to-end from scratch, and building upon pre-trained foundation models. Among those trained from scratch are Chameleon (Team et al., 2024), Show-o (Xie et al., 2025a), VILA-U (Wu et al., 2025d), Janus (Wu et al., 2024), JanusPro (Chen et al., 2025d), JanusFlow (Ma et al., 2024b), and Harmon (Wu et al., 2025c). These systems employ diverse architectures, including autoregressive (AR) and masked autoregressive (MAR) frameworks, to jointly handle both modalities.

The second paradigm leverages pre-trained components, integrating powerful Multimodal Large Language Models (MLLMs) with established diffusion backbones. Notable examples include DreamLLM (Dong et al., 2024), Transfusion (Zhou et al., 2025), MetaQueries (Pan et al., 2025), BLIP3-o (Chen et al., 2025a), UniWorld-V1 (Lin et al., 2025), Qwen-Image (Wu et al., 2025a), and Bagel (Deng et al., 2025). These approaches typically bridge the frozen MLLM and diffusion model using mechanisms like learnable queries or multi-stage training protocols (Pan et al., 2025) to harmonize understanding and generative processes. The resulting synergy of generation and comprehension enables these unified models to tackle a wide spectrum of tasks, including high-fidelity, instruction-guided image editing (Google, 2025a;b).

Despite these significant advances, a fundamental limitation persists across existing unified models. Current training paradigms depend heavily on meticulously curated, large-scale datasets of high-quality image-text pairs to train their generative modules. This reliance on proprietary or difficult-to-acquire data poses a significant barrier to open research and broader community-driven development.

**Masked signal modeling.** Masked signal modeling, pioneered by Masked Autoencoders (MAE) (He et al., 2022), has become a powerful self-supervised learning paradigm. The core prin-

ciple involves training a model to learn robust representations by reconstructing randomly masked portions of an input signal. Initially applied to images, this "mask-and-predict" strategy has been successfully adapted to a diverse range of generative tasks. Notable adaptations include predicting masked visual tokens for non-autoregressive image synthesis (Chang et al., 2022), masking textual conditions to refine guidance in diffusion models (Zhou et al., 2023), leveraging attention mechanisms to generate precise editing masks from user intent (Zou et al., 2024), and improving the data efficiency of Generative Adversarial Network (GAN) training (Huang et al., 2022). The versatility of this approach underscores its potential as a flexible and potent tool for representation learning and generative modeling.

## 3 METHODOLOGY

We propose a novel framework for pre-training a generative model by leveraging a frozen Multimodal Large Language Model (MLLM) with an image-only dataset (see Sec. 3.2), entirely eschewing the need for paired text. Our approach hinges on two key contributions. First, to adapt the MLLM's representations for the generative task without costly fine-tuning, we introduce the ***Residual Query Adapter*** (see Sec. 3.3), a lightweight, parameter-efficient module that refines the visual condition. Second, to prevent the self-conditioning from collapsing to a trivial identity mapping, we employ a ***Masked Image Modeling*** strategy (see Sec. 3.4). This transforms training into a sparse-to-dense reconstruction task, compelling the model to learn a robust and compositional visual prior.

### 3.1 PRELIMINARIES ON DIFFUSION-BASED GENERATIVE MODELS

Diffusion-based generative models transform a simple prior distribution, e.g., a standard Gaussian $\mathcal{N}(\mathbf{0}, \mathbf{I})$, into a complex data distribution by learning to reverse a predefined noise-corruption process. In this paper, we focus on flow matching (FM) models (Lipman et al., 2022), which have demonstrated strong performance in image generation (Xie et al., 2024; Sun et al., 2025).

Flow matching models define a deterministic path from a data point $\mathbf{x}$ to a noise vector $\mathbf{z} \sim \mathcal{N}(\mathbf{0}, \mathbf{I})$ via the interpolation $\mathbf{x}_t = (1 - t) \cdot \mathbf{x} + t \cdot \mathbf{z}$ for $t \in [0, 1]$. A neural network $\boldsymbol{F}_{\boldsymbol{\theta}}(\mathbf{x}_t, t, \mathbf{c})$ is then trained to learn the constant-velocity vector field $\mathbf{z} - \mathbf{x}$ of this path. Formally, given a conditioning signal $\mathbf{c}$, the objective is: $\mathcal{L}(\boldsymbol{\theta}) = \mathbb{E}_{\mathbf{x}, \mathbf{z}, \mathbf{c}, t} \left[ \|\boldsymbol{F}_{\boldsymbol{\theta}}(\mathbf{x}_t, t, \mathbf{c}) - (\mathbf{z} - \mathbf{x})\|_2^2 \right]$.

For generation, one starts with a sample from the prior, $\mathbf{x}_1 \sim \mathcal{N}(\mathbf{0}, \mathbf{I})$, and integrates the learned vector field backward in time from $t = 1$ to $t = 0$. This is achieved by solving the probability flow ordinary differential equation (PF-ODE) (Song et al., 2020): $\frac{\mathrm{d}\mathbf{x}_t}{\mathrm{d}t} = \boldsymbol{F}_{\boldsymbol{\theta}}(\mathbf{x}_t, t, \mathbf{c})$. The solution at $t = 0$ yields the final generated sample $\mathbf{x}_0$.

### 3.2 IMAGE-ONLY PRE-TRAINING VIA SELF-CONDITIONING

We hypothesize that explicit text is merely one possible modality for conveying the high-level semantic information necessary to guide image synthesis. The rich semantic content inherent in an image can itself serve as a sufficient conditioning signal. This principle allows us to design a training paradigm that relies exclusively on an unlabeled image corpus.

Our framework utilizes a pre-trained and frozen MLLM, which we denote as $\boldsymbol{g}$. This MLLM includes a Vision Transformer (ViT) encoder, $\boldsymbol{v}$, for processing visual inputs. To generate an image $\mathbf{x}$, we first derive a conditioning signal directly from $\mathbf{x}$.

**Forming the self-conditioning signal.** Inspired by instruction-following models, we construct the initial condition by combining a generic, fixed textual prompt with the visual features of the image. Let $\mathbf{c}_{\text{aux}} \in \mathbb{R}^{T \times D}$ be the token embeddings for an auxiliary prompt, such as "`Generate an image that is identical to the reference image:`". The ViT encoder $\boldsymbol{v}$ processes the image $\mathbf{x}$ into a sequence of patch embeddings, $\mathbf{c}_{\text{img}} = \boldsymbol{v}(\mathbf{x}) \in \mathbb{R}^{P^2 \times D}$, where $P^2$ is the number of patches and $D$ is the embedding dimension.

The complete conditioning sequence $\mathbf{c}$ is formed by concatenating these two components: $\mathbf{c} = \text{concat}(\mathbf{c}_{\text{aux}}, \mathbf{c}_{\text{img}}) \in \mathbb{R}^{(T + P^2) \times D}$. This sequence is then processed by the frozen MLLM $\boldsymbol{g}$ to produce the final latent condition $\mathbf{h} = \boldsymbol{g}(\mathbf{c})$, which is used to guide the diffusion model $\boldsymbol{F}_{\boldsymbol{\theta}}$.

---

**Algorithm 1** Image-Only Pre-training Framework for UMM Generation

---

**Require:** Image dataset $D$; frozen pre-trained MLLM $g$; frozen ViT encoder $v$; auxiliary prompt embeddings $\mathbf{c}_{\text{aux}}$; mask ratio $r$.
**Require:** Randomly initialized diffusion network $F_{\theta}$ and residual query adapter $q_{\theta}$.
1: **repeat**
2:     Sample image $\mathbf{x} \sim D$, noise $\mathbf{z} \sim \mathcal{N}(\mathbf{0}, \mathbf{I})$, time $t \sim \mathcal{U}(0, 1)$.
3:     Compute noised image: $\mathbf{x}_t = (1 - t) \cdot \mathbf{x} + t \cdot \mathbf{z}$.
4:     Extract image patch embeddings: $\mathbf{c}_{\text{img}} = v(\mathbf{x})$.
5:     Generate random mask $\mathbf{M}$ with masking ratio $r$ and apply it: $\mathbf{c}_{\text{img}} \leftarrow \mathbf{c}_{\text{img}} \odot \mathbf{M}$.
6:     Form the initial condition: $\mathbf{c} = \text{concat}(\mathbf{c}_{\text{aux}}, \mathbf{c}_{\text{img}})$.
7:     Refine condition with residual query adapter: $\mathbf{c} \leftarrow \text{concat}(\mathbf{c}, q_{\theta}(\mathbf{c}))$.
8:     Compute latent condition from frozen MLLM: $\mathbf{h} = g(\mathbf{c})$.
9:     Compute loss: $\mathcal{L}(\theta) = \|F_{\theta}(\mathbf{x}_t, t, \mathbf{h}) - (\mathbf{z} - \mathbf{x})\|_2^2$.
10:    Update trainable parameters $\theta$ using gradients from $\mathcal{L}(\theta)$.
11: **until** convergence

---

### 3.3 RESIDUAL QUERY ADAPTER FOR GENERATIVE ADAPTATION

Directly using the output of a frozen MLLM, $g(\mathbf{c})$, as a condition for the diffusion model yields suboptimal performance (see "Raw" in Fig. 2b). We attribute this to a domain mismatch: representations from an MLLM pre-trained for discriminative or understanding-based tasks are not inherently optimized for the nuanced control required by a generative process.

While fine-tuning the entire MLLM ($g$) could in principle align its representations, this approach is fraught with two major challenges:

(a) the immense computational cost associated with billions of parameters, where e.g. the MLLM in MetaQuery-XL has 7B parameters, versus 0.6B for the diffusion model (Pan et al., 2025).
(b) the risk of catastrophic forgetting, where the powerful, pre-trained capabilities of the MLLM are degraded when fine-tuned on an image-only reconstruction task.

To circumvent these issues, we introduce the **Residual Query Adapter (RQA)**, denoted $q_{\theta}$. The RQA is a lightweight, trainable adapter module designed to preprocess the conditioning signal $\mathbf{c}$ before it enters the MLLM. Specifically, the RQA consists of a single Transformer block (Vaswani et al., 2017) that learns a task-specific transformation. It generates a "residual query" that is appended to the original conditioning sequence: $\mathbf{c} \leftarrow \text{concat}(\mathbf{c}, q_{\theta}(\mathbf{c}))$. The MLLM then processes this refined sequence, $\mathbf{h} = g(\mathbf{c})$. The RQA acts as a learnable "prompt", guiding the frozen MLLM to extract features that are more salient for the downstream generative task without modifying any of the MLLM's original weights.

This parameter-efficient approach effectively adapts the MLLM for generation at a fraction of the computational cost. The efficacy of the RQA is empirically validated in Fig. 2b and Sec. 4.4.

### 3.4 MASKED IMAGE MODELING FOR SPARSE RECONSTRUCTING

A key characteristic of conventional text-to-image training is the inherent sparsity of supervision: a short textual description provides only a high-level, incomplete specification of the corresponding image (Xie et al., 2024; Labs et al., 2025b). This forces the model to learn a compositional understanding of scenes and objects to fill in the missing details. In contrast, our self-conditioning approach provides a dense, complete representation of the target image, which can encourage the model to learn a trivial identity mapping rather than a meaningful generative prior.

To emulate the benefits of sparse supervision, we introduce a **Masked Image Modeling** strategy inspired by masked autoencoders (He et al., 2022). During training, we randomly mask a fraction of the image patch tokens $\mathbf{c}_{\text{img}}$ with a masking ratio $r \in [0, 1]$. This is implemented by element-wise multiplication with a binary mask $\mathbf{M} \in \{0, 1\}^{P^2 \times D}$, where entries are drawn from a Bernoulli distribution, and applying it via element-wise multiplication: $\mathbf{c}_{\text{img}} \leftarrow \mathbf{c}_{\text{img}} \odot \mathbf{M}$. This simple yet effective technique transforms the training objective from dense reconstruction to a more challenging sparse-to-dense task. The model is forced to infer the content of the masked patches from the visible ones, promoting the learning of robust, context-aware visual representations. As shown

in our experiments (see Fig. 2b and Sec. 4.4), this significantly improves generation quality. Our complete training procedure is detailed in Alg. 1 and Fig. 2.

## 4 EXPERIMENT

We conduct a comprehensive experiments to validate the efficacy of our proposed framework, IOMM. Our evaluation is designed to systematically assess its performance in text-to-image generation, analyze the impact of different training data compositions, and ablate its core architectural components.

### 4.1 EXPERIMENTAL SETTING

**Datasets.** Our pre-training corpus comprises the Megalith-10M (Matsubara & Team, 2024) and text-to-image-2M (He & contributors, 2024) datasets. For the fine-tuning stage, we leverage a curated collection of high-quality, instruction-following datasets, namely BLIP3-o-60K (Chen et al., 2025a), Echo-4o-Image (Ye et al., 2025a), and ShareGPT-4o-Image (Chen et al., 2025c). All images undergo a standardized preprocessing pipeline: we apply a central crop and resize them to a resolution of either $512 \times 512$ or $1024 \times 1024$. The processed images are then encoded into a latent representation using a pre-trained DC-AE (*f32c32*) model (Chen et al., 2024c; Xie et al., 2024).

**Neural network architectures.** The core of our generative model is the widely-adopted Multi-Modal Diffusion Transformer (MM-DiT) architecture (Esser et al., 2024), as implemented in the FLUX framework (Labs et al., 2025a). This architecture is notable for its use of independent attention mechanisms for image and text modalities, which facilitates robust cross-modal fusion. We instantiate two model variants to investigate scaling properties: IOMM-B and IOMM-L, which are built upon MM-DiT backbones with 1.6B and 2.7B parameters, respectively. For the auxiliary MLLM component, we employ the compact yet powerful InternVL3-2B model (Zhu et al., 2025) as a frozen feature extractor. Its modest size significantly reduces the overall computational footprint without compromising feature quality.

**Implementation and evaluation.** We implement our framework in PyTorch (Paszke, 2019) and utilize the AdamW optimizer (Loshchilov & Hutter, 2017) for training. Adhering to established practices in generative modeling (Yu et al., 2024; Ma et al., 2024a), we maintain an exponential moving average (EMA) of the model weights with a decay rate of 0.999. All reported results are derived from the EMA model weights to ensure stability and improved performance. For evaluation, we follow standard protocols established in prior works (Pan et al., 2025; Chen et al., 2025d; Esser et al., 2024). To assess generative quality and text-image alignment, we employ a suite of comprehensive benchmarks: GenEval (Ghosh et al., 2023), DPG-Bench (Hu et al., 2024), and WISE (Niu et al., 2025). The image editing capabilities of our model are specifically evaluated using the ImgEdit-Bench (Ye et al., 2025b). Further details regarding hyperparameters and the training infrastructure are available in App. B.

### 4.2 PERFORMANCE ON TEXT-TO-IMAGE GENERATION

We benchmark IOMM against SOTA models in Tab. 1. Our base model, IOMM-B (512px) built on a 1.6B generative backbone, achieves a new SOTA score of 0.89 on GenEval. Notably, this performance surpasses strong baselines like BAGEL (0.88) and BLIP3-o-8B*(0.84, trained with an extra 30M proprietary image-text pairs), despite IOMM being trained exclusively on public datasets and with remarkable efficiency (1,050 H800 GPU hours). Furthermore, IOMM-B attains a competitive score of 0.55 on the WISE benchmark, demonstrating that our approach effectively preserves world knowledge without degradation. Qualitative results in Fig. 3 showcase our model's strong compositional abilities.

**Analysis of model scaling.** The lower performance of our larger IOMM-L model is an artifact of constrained training resources; it was trained for half the epochs of IOMM-B. When controlling for training duration (5 epochs), IOMM-L outperforms IOMM-B (0.87 vs. 0.86 on GenEval), confirming a positive scaling trend and suggesting potential for further gains with continued training.

### 4.3 IMPACT OF PRE-TRAINING AND FINE-TUNING DATA COMPOSITION

We investigate the impact of data composition during the pre-training and fine-tuning stages. We define three distinct data types: (a) image-only, (b) text-image pairs, and (c) a mixture of both. This section presents a systematic ablation study on the six possible combinations of these data types across the two stages, focusing on their efficacy for text-to-image generation.

Table 1: **Quantitative comparison on text-to-image generation benchmarks.** The (↑) symbol indicates that higher scores are better. [†]Results obtained using rewritten prompts from the original GenEval benchmark. [*]Indicates the model was trained on an additional 30M proprietary image-text pairs.

| METHOD | GenEval | | | | | | | DPGBench (↑) | WISE (↑) |
|---|---|---|---|---|---|---|---|---|---|
| | Single Obj. | Two Obj. | Counting | Colors | Position | Color Attri. | Overall (↑) | | |
| **Gen. Only** | | | | | | | | | |
| SDv1.5 (Rombach et al., 2022) | 0.97 | 0.38 | 0.35 | 0.76 | 0.04 | 0.06 | 0.43 | 63.18 | 0.32 |
| SDv2.1 (Rombach et al., 2022) | 0.98 | 0.51 | 0.44 | 0.85 | 0.07 | 0.17 | 0.50 | - | 0.32 |
| SD3-Medium (Esser et al., 2024) | 0.99 | 0.94 | 0.72 | 0.89 | 0.33 | 0.60 | 0.74 | 84.08 | 0.42 |
| SDXL (Podell et al., 2024) | 0.98 | 0.74 | 0.39 | 0.85 | 0.15 | 0.23 | 0.55 | 74.65 | 0.43 |
| PixArt-α (Chen et al., 2024b) | 0.98 | 0.50 | 0.44 | 0.80 | 0.08 | 0.07 | 0.48 | 71.11 | 0.47 |
| DALL-E 2 (Ramesh et al., 2022) | 0.94 | 0.66 | 0.49 | 0.77 | 0.10 | 0.19 | 0.52 | - | - |
| DALL-E 3 (Betker et al., 2023) | 0.96 | 0.87 | 0.47 | 0.83 | 0.43 | 0.45 | 0.67 | 83.50 | - |
| **Unified Models** | | | | | | | | | |
| Chameleon (Team et al., 2024) | - | - | - | - | - | - | 0.39 | - | - |
| Show-o (Xie et al., 2025a) | 0.98 | 0.80 | 0.66 | 0.84 | 0.31 | 0.50 | 0.68 | - | 0.35 |
| Show-o2-7B (Xie et al., 2025b) | 1.00 | 0.87 | 0.58 | 0.92 | 0.52 | 0.62 | 0.76† | 86.14 | 0.39 |
| Janus (Wu et al., 2024) | 0.97 | 0.68 | 0.30 | 0.84 | 0.46 | 0.42 | 0.61 | 79.68 | 0.23 |
| JanusFlow (Ma et al., 2024b) | 0.97 | 0.59 | 0.45 | 0.83 | 0.53 | 0.42 | 0.63 | 80.09 | - |
| Janus-Pro-1B (Chen et al., 2025d) | 0.98 | 0.82 | 0.51 | 0.89 | 0.65 | 0.56 | 0.73 | 82.63 | 0.26 |
| Janus-Pro-7B (Chen et al., 2025d) | 0.99 | 0.89 | 0.59 | 0.90 | 0.79 | 0.66 | 0.80 | 84.19 | 0.35 |
| MetaQuery-B (Pan et al., 2025) | - | - | - | - | - | - | 0.74† | 80.04 | 0.46 |
| MetaQuery-L (Pan et al., 2025) | - | - | - | - | - | - | 0.78† | 81.10 | 0.55 |
| MetaQuery-XL (Pan et al., 2025) | - | - | - | - | - | - | 0.80† | 82.05 | 0.55 |
| BLIP3-o-4B (Chen et al., 2025a) | - | - | - | - | - | - | 0.81 | 79.36 | 0.50 |
| BLIP3-o-8B* (Chen et al., 2025a) | - | - | - | - | - | - | 0.84 | 81.60 | 0.62 |
| BAGEL-7B (Deng et al., 2025) | 0.98 | 0.95 | 0.84 | 0.95 | 0.78 | 0.77 | 0.88† | - | 0.52 |
| **Ours** | | | | | | | | | |
| IOMM-B 512 | 0.99 | 0.92 | 0.83 | 0.94 | 0.91 | 0.75 | 0.89 | 82.95 | 0.55 |
| IOMM-B 1024 | 0.99 | 0.91 | 0.75 | 0.93 | 0.88 | 0.75 | 0.87 | 80.71 | 0.50 |
| IOMM-L 512 | 0.99 | 0.91 | 0.82 | 0.94 | 0.85 | 0.72 | 0.87 | 76.09 | 0.53 |
| IOMM-L 1024 | 1.00 | 0.91 | 0.71 | 0.92 | 0.78 | 0.78 | 0.85 | 72.26 | 0.48 |

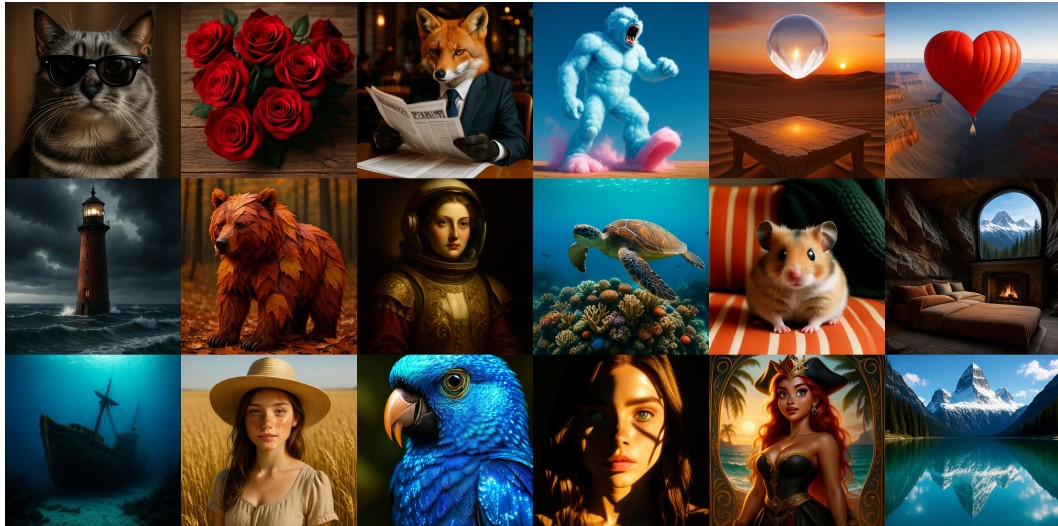

Figure 3: **Our model demonstrates proficiency in handling complex prompts involving multiple objects, spatial relationships, and nuanced attributes.** Corresponding prompts are provided in App. C.7.

**The role of pre-training data.** We first compare models pre-trained on image-only data versus those pre-trained on text-image pairs. As illustrated in Fig. 4 and Fig. 1d, the image-only pre-trained model consistently achieves superior or comparable performance to its text-image pair counterpart, irrespective of the fine-tuning data composition.

**The role of fine-tuning data.** Next, we analyze the effect of the fine-tuning data composition. Beyond using image-only or text-image pair data exclusively, we explore a mixed-data strategy. Remarkably, Fig. 1d reveals that for models pre-trained under **both** paradigms, fine-tuning with the mixed data yields the highest performance on GenEval. Conversely, fine-tuning with image-only data consistently results in the lowest scores.

The training dynamics, detailed in Fig. 4d, offer further insight. Although the mixed-data strategy initially lags behind the text-image pair approach, it demonstrates superior scaling properties and ultimately surpasses it with extended training. In stark contrast, the image-only fine-tuning approach performs poorly throughout the entire process. We hypothesize that this is because

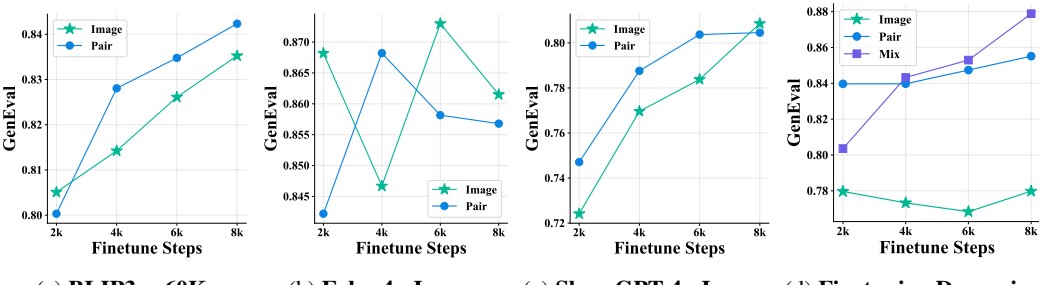

|  (a) **BLIP3-o-60K.** | (b) **Echo-4o-Image.** | (c) **ShareGPT-4o-Image.** | (d) **Finetuning Dynamics.** |

Figure 4: **Analysis of different data paradigms.** (a-c) Performance comparison of fine-tuning with three data compositions (image-only, text-image pairs, mixed) on distinct datasets. (d) Fine-tuning performance scaling over training steps, illustrating the superior long-term potential of the mixed-data approach compared to using text-image pairs alone.

Table 2: **Evaluating different fine-tuning strategies on various open-source UMMs.** The notation A $\oplus$ B denotes applying fine-tuning method B to a pre-trained model A. The symbols $\downarrow$/$\uparrow$ indicate the performance change relative to the baseline pre-trained model.

| METHOD | Res. | NFE | Single Obj. | Two Obj. | Counting | GenEval Colors | Position | Color Attri. | Overall ($\uparrow$) | WISE ($\uparrow$) |
|---|---|---|---|---|---|---|---|---|---|---|
| OpenUni-L (Wu et al., 2025b) | 512 | 20×2 | 0.99 | 0.91 | 0.77 | 0.90 | 0.75 | 0.76 | 0.85 | 0.52 |
| $\oplus$Image finetuning | 512 | 20×2 | $1.00^{\uparrow 0.01}$ | $0.98^{\uparrow 0.07}$ | $0.22^{\downarrow 0.55}$ | $0.91^{\uparrow 0.01}$ | $0.60^{\downarrow 0.15}$ | $0.77^{\uparrow 0.01}$ | $0.74^{\downarrow 0.11}$ | $0.49^{\downarrow 0.03}$ |
| $\oplus$Pair finetuning | 512 | 20×2 | 0.99 | $0.94^{\uparrow 0.03}$ | $0.82^{\uparrow 0.05}$ | $0.91^{\uparrow 0.01}$ | $0.85^{\uparrow 0.10}$ | 0.76 | $0.88^{\uparrow 0.03}$ | $0.62^{\uparrow 0.10}$ |
| $\oplus$Mix finetuning | 512 | 20×2 | 0.99 | 0.91 | $0.78^{\uparrow 0.01}$ | $0.93^{\uparrow 0.03}$ | $0.87^{\uparrow 0.12}$ | $0.78^{\uparrow 0.02}$ | $0.88^{\uparrow 0.03}$ | $0.59^{\uparrow 0.07}$ |
| Qwen-Image (Wu et al., 2025a) | 512 | 50×2 | 0.99 | 0.91 | 0.87 | 0.88 | 0.73 | 0.74 | 0.85 | - |
| $\oplus$Image finetuning | 512 | 50×2 | $0.55^{\downarrow 0.44}$ | $0.51^{\downarrow 0.40}$ | $0.38^{\downarrow 0.49}$ | $0.43^{\downarrow 0.45}$ | $0.30^{\downarrow 0.43}$ | $0.37^{\downarrow 0.37}$ | $0.42^{\downarrow 0.43}$ | 0.41 |
| $\oplus$Pair finetuning | 512 | 50×2 | $1.00^{\uparrow 0.01}$ | $0.93^{\uparrow 0.02}$ | $0.88^{\uparrow 0.01}$ | $0.91^{\uparrow 0.03}$ | $0.82^{\uparrow 0.09}$ | $0.75^{\uparrow 0.01}$ | $0.88^{\uparrow 0.03}$ | 0.63 |
| $\oplus$Mix finetuning | 512 | 50×2 | $1.00^{\uparrow 0.01}$ | $0.92^{\uparrow 0.01}$ | 0.87 | $0.91^{\uparrow 0.03}$ | $0.82^{\uparrow 0.09}$ | $0.79^{\uparrow 0.05}$ | $0.89^{\uparrow 0.04}$ | 0.63 |
| Qwen-Image (Wu et al., 2025a) | 1024 | 50×2 | 0.99 | 0.93 | 0.88 | 0.90 | 0.77 | 0.74 | 0.87 | 0.62 |
| $\oplus$Image finetuning | 1024 | 50×2 | $0.54^{\downarrow 0.45}$ | $0.61^{\downarrow 0.32}$ | $0.47^{\downarrow 0.41}$ | $0.47^{\downarrow 0.43}$ | $0.28^{\downarrow 0.49}$ | $0.47^{\downarrow 0.27}$ | $0.47^{\downarrow 0.40}$ | $0.35^{\downarrow 0.27}$ |
| $\oplus$Pair finetuning | 1024 | 50×2 | $1.00^{\uparrow 0.01}$ | $0.93^{\uparrow 0.01}$ | 0.88 | $0.91^{\uparrow 0.01}$ | $0.82^{\uparrow 0.05}$ | $0.75^{\uparrow 0.01}$ | $0.88^{\uparrow 0.01}$ | $0.63^{\uparrow 0.01}$ |
| $\oplus$Mix finetuning | 1024 | 50×2 | 0.99 | $0.92^{\downarrow 0.01}$ | $0.90^{\uparrow 0.02}$ | $0.91^{\uparrow 0.01}$ | $0.81^{\uparrow 0.04}$ | $0.80^{\uparrow 0.06}$ | $0.89^{\uparrow 0.02}$ | $0.63^{\uparrow 0.01}$ |

text-to-image generation demands strong prompt-following capabilities, which image-only data cannot explicitly provide.

**Generalization to open-source UMMs.** To validate the generalizability of our findings, we apply our fine-tuning strategies to prominent open-source Unified Multimodal Models (UMMs): OpenUni-L-3.6B (Wu et al., 2025b) and Qwen-Image-20B (Wu et al., 2025a). For the larger Qwen-Image model, we employ LoRA (Hu et al., 2022) (with $r = 64$ and $\alpha = 64$) for computational efficiency. The results, summarized in Tab. 2, corroborate our primary conclusion: the mixed-data fine-tuning approach consistently outperforms the other strategies on GenEval. For instance, it improves the GenEval score of OpenUni-L from a baseline of $0.85$ to $0.88$. Even for the powerful Qwen-Image model, this strategy yields notable gains, increasing scores from $0.85$ to $0.89$ (512 resolution) and $0.87$ to $0.89$ (1024 resolution).

Beyond generation quality, we evaluate world knowledge and reasoning using the WISE benchmark. As shown in the final column of Tab. 2, both text-image pair and mixed-data fine-tuning provide a substantial performance uplift for OpenUni-L (up to $0.10$) and a modest improvement for Qwen-Image ($0.01$). In contrast, fine-tuning with image-only data proves detrimental across nearly all scenarios, significantly impairing the models' prompt-following abilityan effect particularly pronounced in larger models (see App. C.6 for a detailed analysis).

**Emergent image editing capabilities.** A surprising and significant finding is the emergence of strong image editing capabilities. Tab. 3 demonstrates that our model, when pre-trained on image-only data, achieves competitive performance on the ImgEdit-Bench benchmark. Crucially, this is accomplished in a *zero-shot setting*, without any fine-tuning on task-specific editing data. This training-free approach not only surpasses the performance of the same model pre-trained on text-image pairs but also outperforms several strong baselines like UltraEdit (Zhao et al., 2024) that are explicitly trained on editing datasets.

### 4.4 ABLATION STUDIES ON KEY COMPONENTS OF IOMM

Unless specified otherwise, all experiments in this section are conducted using the IOMM-B model pre-trained exclusively on image-only data.

Table 3: **Image editing benchmark results.** Methods highlighted in `red` are trained on specific editing datasets. Our IOMM, highlighted in `blue`, is evaluated in a `training-free` setting without any using on editing data.

| METHOD | Add | Adjust | Extract | Replace | Remove | Background | Style | Hybrid | Action | Overall (↑) |
|---|---|---|---|---|---|---|---|---|---|---|
| | | | | | **ImgEdit-Bench** | | | | | |
| | | | | | **Trained with editing data** | | | | | |
| MagicBrush (Zhang et al., 2023) | 2.84 | 1.58 | 1.51 | 1.97 | 1.58 | 1.75 | 2.38 | 1.62 | 1.22 | 1.90 |
| Instruct-Pix2Pix (Brooks et al., 2023) | 2.45 | 1.83 | 1.44 | 2.01 | 1.50 | 1.44 | 3.55 | 1.20 | 1.46 | 1.88 |
| AnyEdit (Yu et al., 2025) | 3.18 | 2.95 | 1.88 | 2.47 | 2.23 | 2.24 | 2.85 | 1.56 | 2.65 | 2.45 |
| UltraEdit (Zhao et al., 2024) | 3.44 | 2.81 | 2.13 | 2.96 | 1.45 | 2.83 | 3.76 | 1.91 | 2.98 | 2.70 |
| OmniGen (Xiao et al., 2025) | 3.47 | 3.04 | 1.71 | 2.94 | 2.43 | 3.21 | 4.19 | 2.24 | 3.38 | 2.96 |
| ICEdit (Zhang et al., 2025) | 3.58 | 3.39 | 1.73 | 2.93 | 3.15 | 3.08 | 3.84 | 2.04 | 3.68 | 3.05 |
| Step1X-Edit (Liu et al., 2025) | 3.88 | 3.14 | 1.76 | 3.40 | 2.41 | 3.16 | 4.63 | 2.64 | 2.52 | 3.06 |
| BAGEL (Deng et al., 2025) | 3.56 | 3.31 | 1.70 | 3.3 | 2.62 | 3.24 | 4.49 | 2.38 | 4.17 | 3.20 |
| | | | | | **Ours (zero-shot)** | | | | | |
| IOMM-B (text-image pair pre-trained) | 3.18 | 2.17 | 1.92 | 2.70 | 1.17 | 3.36 | 4.39 | 1.49 | 3.14 | 2.61 |
| IOMM-B (image-only pre-trained) | 3.84 | 2.37 | 2.12 | 2.60 | 1.30 | 3.14 | 4.41 | 1.80 | 3.78 | 2.82 |

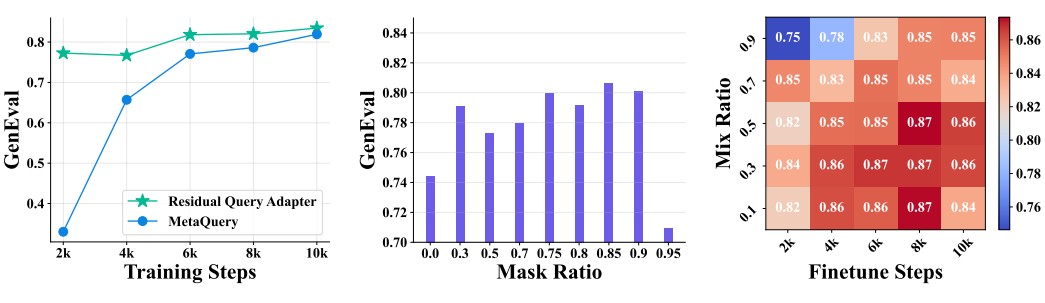

(a) **Residual query adapter.**     (b) **Various mask ratio.**     (c) **Various mix ratio of data.**

Figure 5: **Ablation studies of key components in IOMM.** These experiments analyze the impact of our primary design choices: (a) the residual query adapter, (b) the mask ratio for sparse reconstruction, and (c) the data mixture ratio during fine-tuning.

**Efficacy of the residual query adapter.** To further validate the efficacy of our proposed residual query adapter, we compare it against a strong baseline, MetaQuery (Pan et al., 2025), trained on identical data. The results, depicted in Fig. 5a, clearly demonstrate that our approach achieves a significantly faster convergence rate.

**Impact of image token mask ratio.** We investigate the impact of the mask ratio for image tokens, a key parameter in our sparse reconstruction objective. As shown in Fig. 5b, performance improves as the ratio increases, peaking at an impressive 0.81 GenEval score. This result validates the effectiveness of our learning paradigm. However, an excessively high ratio (e.g., 0.95) leads to a sharp performance degradation (a drop to 0.71), likely due to significant information loss that impairs the training guidance for the generation process.

**Influence of data mixture ratio.** We examine the effect of varying the proportion of image-only data versus text-image pairs during the fine-tuning stage. A mix ratio of 1.0 corresponds to pure image-only data, while 0.0 signifies pure text-image pairs. Fig. 5c reveals that performance initially increases with the mix ratio, reaching its optimum at 0.5. Furthermore, an optimal ratio of approximately 0.5 not only yields the best results but also demonstrates greater training stability, whereas lower ratios are prone to performance volatility in the later stages of fine-tuning.

## 5 CONCLUSION AND LIMITATIONS

We have introduced IOMM, a novel end-to-end framework for training unified models from scratch using only image-only data. Our framework is highly modular, allowing for seamless integration with existing MLLMs and diffusion models to adapt them into powerful unified models with minimal fine-tuning effort. Furthermore, we demonstrate that our proposed mixed-data fine-tuning strategy consistently enhances the performance of existing UMMs. Extensive experimentation validates the SOTA performance, effectiveness, and efficiency of IOMM. Additional details on experimental settings and results are provided in App. B.

## 6 ETHICS STATEMENT

This research adheres to the *ICLR Code of Ethics* and is committed to the principles of responsible and transparent scientific inquiry. The study involves no human participants, personal or sensitive data, or any activities requiring approval from an institutional ethics review board. All datasets used are publicly accessible under appropriate licenses, with proper attribution given to their original sources. To promote openness and reproducibility, we provide our implementation code and experimental settings for verification and further development by the research community. We also declare that no conflicts of interest or external funding have influenced the design, execution, or presentation of this work.

## 7 REPRODUCIBILITY STATEMENT

Comprehensive details regarding the datasets, model architectures, optimization settings, and training procedures are provided in Sec. 4.1 of the main paper and in App. B. These materials are designed to facilitate the reliable and transparent reproduction of our results. Additionally, our source code will be made publicly available upon acceptance of the paper.

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

CONTENTS

## A   UTILIZATION OF LARGE LANGUAGE MODELS (LLMS)

In this study, Large Language Models (LLMs) are employed at the sentence level to assist in linguistic refinement. Their use was strictly confined to improving grammatical accuracy and overall readability of the manuscript. All research concepts, methodological designs, experimental processes, and analytical findings remain entirely original and have been solely contributed by the authors.

## B   DETAILED EXPERIMENTAL SETTINGS

This section elaborates on the experimental setup, including all relevant hyperparameter choices.

### B.1   PRETRAINING SETTINGS

The results presented in Tab. 1 are derived using the pretraining configurations outlined in Tab. 4. Due to computational resource constraints, Exponential Moving Average (EMA) decay was not applied during the training of IOMM-L. Both models were pretrained on the Megalith-10M (Matsubara & Team, 2024) and text-to-image-2M (He & contributors, 2024) datasets, totaling approximately 11 million images. Each image was centrally cropped and resized to $512 \times 512$. Subsequently, a Deep Compression Autoencoder (DC-AE, *f32c32*) (Chen et al., 2024c) was employed to map the images into the latent space. Notably, since neither dataset provides images at a resolution of $1024 \times 1024$, we did not deploy high-resolution pretraining.

Table 4: **Pretraining settings.**

| METHOD | IOMM-B | IOMM-L |
|---|---|---|
| Optimization | | |
| Optimizer | AdamW | |
| $\beta$ | $(0.9, 0.95)$ | |
| Learning rate | 1e-4 | |
| Max gradient norm | 1.0 | |
| Weight decay | 0.0 | |
| Training Configuration | | |
| Generative Model Size | 1.6B | 2.7B |
| Training data type | Image-only | Image-only |
| EMA decay | 0.999 | - |
| Global batch size | 1024 | 512 |
| Epochs | 10 | 5 |
| Image token mask ratio $r$ | 0.85 | 0.85 |

### B.2   FINETUNING SETTINGS

We fine-tuned the two models at resolutions of 512 and 1024, respectively, using the pretraining settings specified in Tab. 4. The fine-tuning datasets include BLIP3o-60K (Chen et al., 2025a), Echo-4o-Image (Ye et al., 2025a), and ShareGPT-4o-Image (Chen et al., 2025c), collectively comprising approximately 210,000 high-resolution images. All images in these datasets are at $1024 \times 1024$ resolution. For fine-tuning at both 512 and 1024 resolutions, we applied central cropping to resize images to the target resolution and utilized a Deep Compression Autoencoder (DC-AE, *f32c32*) (Chen et al., 2024c) to map images into the latent space. The IOMM-L-1024 model was fine-tuned using Distributed Data Parallel (DDP) across two GPU nodes to maintain the same batch size as IOMM-B-1024.

### B.3   UMM FINETUNING SETTINGS

The results presented in Tab. 2 were obtained using the fine-tuning configurations specified in Tab. 6. For OpenUni-L, we performed full fine-tuning on both the connector module and the generative model. In contrast, for Qwen-Image-20B, we applied Low-Rank Adaptation (LoRA) (Hu et al., 2022) to fine-tune the model. Both models utilized a frozen understanding module. Additionally,

Table 5: **Finetuning settings.**

| METHOD | IOMM-B | | IOMM-L | |
|---|---|---|---|---|
| **Resolution** | 512 | 1024 | 512 | 1024 |
| Optimization | | | | |
| Optimizer | AdamW | | AdamW | |
| $\beta$ | $(0.9, 0.95)$ | | $(0.9, 0.95)$ | |
| Learning rate | 1e-4 | | 1e-4 | |
| Max gradient norm | 1.0 | | 1.0 | |
| Weight decay | 0.0 | | 0.0 | |
| Generative Model Size | 1.6B | 1.6B | 2.7B | 2.7B |
| Training Configuration | | | | |
| Training data type | Mix | Mix | Mix | Mix |
| EMA decay | 0.999 | 0.999 | - | - |
| Global batch size | 256 | 96 | 256 | 96 |
| Epochs | 25 | 10 | 25 | 10 |
| Image token mask ratio $r$ | 0.85 | 0.85 | 0.85 | 0.85 |
| Mix ratio $\lambda$ | 0.5 | 0.5 | 0.5 | 0.5 |

due to computational constraints, Exponential Moving Average (EMA) decay was not implemented for Qwen-Image-20B.

Table 6: **UMM finetuning settings.**

| METHOD | OpenUni-L | Qwen-Image-20B |
|---|---|---|
| Optimization | | |
| Optimizer | AdamW | AdamW |
| $\beta$ | $(0.9, 0.95)$ | $(0.9, 0.95)$ |
| Learning rate | 1e-4 | 1e-4 |
| Max gradient norm | 1.0 | 1.0 |
| Weight decay | 0.0 | 0.0 |
| Training Configuration | | |
| Training data type | Mix/Image-only/Pair | Mix/Image-only/Pair |
| EMA decay | 0.999 | - |
| Global batch size | 256 | 48 |
| Epochs | 12 | 5 |
| Image token mask ratio $r$ | 0.85 | 0.85 |
| Mix ratio $\lambda$ | 0.5 | 0.5 |
| LoRA Configuration | | |
| LoRA rank | - | 64 |
| LoRA alpha | - | 64 |
| LoRA dropout | - | 0.0 |

## C MORE RESULTS

### C.1 DPGBENCH EVALUATION RESULTS

The Tab. 7 shows the detailed results of the DPGBench evaluation shown in Tab. 1.

### C.2 WISE EVALUATION RESULTS

The Tab. 8 shows the detailed results of the WISE evaluation shown in Tab. 1.

Table 7: **DPGBench evaluation results.** Notation A⊕B denotes the result obtained by combining methods A and B. ↓/↑ indicate a decrease/increase, respectively, in the metric compared to the baseline performance of the pretrained models. Here BLIP3-o-8B* donates the model that is trained with an 30 million proprietary data.

| METHOD | Global | Entity | Attribute | Relation | Other | Overall |
|---|---|---|---|---|---|---|
| Gen. Only | | | | | | |
| SDv1.5 (Rombach et al., 2022) | 74.63 | 74.23 | 75.39 | 73.49 | 67.81 | 63.18 |
| SD3-Medium (Esser et al., 2024) | 87.90 | 91.01 | 88.83 | 80.70 | 88.68 | 84.08 |
| SDXL (Podell et al., 2024) | 83.27 | 82.43 | 80.91 | 86.76 | 80.41 | 74.65 |
| PixArt-$\alpha$ (Chen et al., 2024b) | 74.97 | 79.32 | 78.60 | 82.57 | 76.96 | 71.11 |
| FLUX.1-dev (BlackForest, 2024) | 74.35 | 90.00 | 88.96 | 90.87 | 88.33 | 83.84 |
| Unified Models | | | | | | |
| Janus (Wu et al., 2024) | 82.33 | 87.38 | 87.70 | 85.46 | 86.41 | 79.68 |
| Janus-Pro-1B (Chen et al., 2025d) | 87.58 | 88.63 | 88.17 | 88.98 | 88.30 | 82.63 |
| Janus-Pro-7B (Chen et al., 2025d) | 86.90 | 88.90 | 89.40 | 89.32 | 89.48 | 84.19 |
| MetaQuery-B (Pan et al., 2025) | - | - | - | - | - | 80.04 |
| MetaQuery-L (Pan et al., 2025) | - | - | - | - | - | 81.10 |
| MetaQuery-XL (Pan et al., 2025) | - | - | - | - | - | 82.05 |
| BLIP3-o-4B (Chen et al., 2025a) | - | - | - | - | - | 79.36 |
| BLIP3-o-8B* (Chen et al., 2025a) | - | - | - | - | - | 81.60 |
| Ours | | | | | | |
| IOMM-B 512 | 91.33 | 89.39 | 90.07 | 86.89 | 87.78 | 82.95 |
| IOMM-B 1024 | 86.20 | 88.39 | 87.69 | 90.11 | 87.05 | 80.71 |
| IOMM-L 512 | 83.28 | 83.61 | 84.69 | 83.46 | 79.83 | 76.09 |
| IOMM-L 1024 | 79.27 | 82.00 | 80.93 | 82.81 | 78.68 | 72.26 |

Table 8: **WISE evaluation results.** Notation A⊕B denotes the result obtained by combining methods A and B. ↓/↑ indicate a decrease/increase, respectively, in the metric compared to the baseline performance of the pretrained models. Here BLIP3-o-8B* donates the model that is trained with an 30 million proprietary data.

| METHOD | Cultural | Time | Space | Biology | Physics | Chemistry | Overall |
|---|---|---|---|---|---|---|---|
| Gen. Only | | | | | | | |
| SDv1.5 (Rombach et al., 2022) | 0.34 | 0.35 | 0.32 | 0.28 | 0.29 | 0.21 | 0.32 |
| SDv2.1 (Rombach et al., 2022) | 0.30 | 0.38 | 0.35 | 0.33 | 0.34 | 0.21 | 0.32 |
| SD3-Medium (Esser et al., 2024) | 0.42 | 0.44 | 0.48 | 0.39 | 0.47 | 0.29 | 0.42 |
| SDXL (Podell et al., 2024) | 0.43 | 0.48 | 0.47 | 0.44 | 0.45 | 0.27 | 0.43 |
| SD3.5-Large (Esser et al., 2024) | 0.44 | 0.50 | 0.58 | 0.44 | 0.52 | 0.31 | 0.46 |
| PixArt-$\alpha$ (Chen et al., 2024b) | 0.45 | 0.50 | 0.48 | 0.49 | 0.56 | 0.34 | 0.47 |
| FLUX.1-dev (Chen et al., 2024b) | 0.48 | 0.58 | 0.62 | 0.42 | 0.51 | 0.35 | 0.50 |
| Unified Models | | | | | | | |
| Show-o (Xie et al., 2025a) | 0.28 | 0.40 | 0.48 | 0.30 | 0.46 | 0.30 | 0.35 |
| Janus (Wu et al., 2024) | 0.16 | 0.26 | 0.35 | 0.28 | 0.30 | 0.14 | 0.23 |
| Janus-Pro-1B (Chen et al., 2025d) | 0.20 | 0.28 | 0.45 | 0.24 | 0.32 | 0.16 | 0.26 |
| Janus-Pro-7B (Chen et al., 2025d) | 0.30 | 0.37 | 0.49 | 0.36 | 0.42 | 0.26 | 0.35 |
| MetaQuery-B (Pan et al., 2025) | 0.44 | 0.49 | 0.58 | 0.41 | 0.49 | 0.34 | 0.46 |
| MetaQuery-L (Pan et al., 2025) | 0.56 | 0.57 | 0.62 | 0.48 | 0.63 | 0.42 | 0.55 |
| MetaQuery-XL (Pan et al., 2025) | 0.56 | 0.55 | 0.62 | 0.49 | 0.63 | 0.41 | 0.55 |
| BAGEL (Deng et al., 2025) | 0.44 | 0.55 | 0.68 | 0.44 | 0.60 | 0.39 | 0.52 |
| BLIP3-o-4B (Chen et al., 2025a) | - | - | - | - | - | - | 0.50 |
| BLIP3-o-8B* (Chen et al., 2025a) | - | - | - | - | - | - | 0.62 |
| Ours: FLUX pretrained from scratch | | | | | | | |
| IOMM-B 512 | 0.50 | 0.56 | 0.66 | 0.49 | 0.72 | 0.46 | 0.55 |
| IOMM-B 1024 | 0.44 | 0.50 | 0.64 | 0.46 | 0.63 | 0.43 | 0.50 |
| IOMM-L 512 | 0.48 | 0.56 | 0.63 | 0.49 | 0.64 | 0.51 | 0.53 |
| IOMM-L 1024 | 0.44 | 0.48 | 0.59 | 0.43 | 0.58 | 0.44 | 0.48 |

## C.3 DIFFERENT TRAINING RECIPE

The results presented in Tab. 9 correspond to the training configurations depicted in Fig. 1c. All models underwent approximately 5 epochs of pretraining on a dataset comprising 11 million images, followed by 10 epochs of fine-tuning on a dataset of 210,000 images. Notably, the model pretrained exclusively on image-only data and fine-tuned on a mixed data achieved superior performance across most metrics in the GenEval benchmark.

Table 9: **Training recipe comparison.** The GenEval score of the models pretrained with different training recipes. **Bold** denotes the best performance and underline denotes the second best performance.

| Finetuning Recipe | Single Obj. | Two Obj. | Counting | Colors | Position | Color Attri. | Overall (↑) |
|---|---|---|---|---|---|---|---|
| **Pretrained with Text-Image Pair Data** | | | | | | | |
| Image | **1.00** | **0.95** | 0.63 | 0.87 | 0.50 | 0.72 | 0.78 |
| Pair | 0.99 | 0.92 | 0.76 | 0.91 | 0.87 | 0.69 | 0.86 |
| Mix | 0.99 | 0.91 | 0.80 | 0.92 | 0.90 | 0.75 | 0.88 |
| **Pretrained with Image-Only Data** | | | | | | | |
| Image | 0.99 | 0.84 | 0.24 | 0.75 | 0.37 | 0.45 | 0.61 |
| Pair | 0.99 | 0.91 | 0.77 | 0.93 | 0.87 | 0.75 | 0.87 |
| Mix | 0.99 | 0.92 | **0.83** | **0.94** | **0.91** | 0.75 | **0.89** |

## C.4 IMAGE EDITING RESULTS

Fig. 6 compares the image editing capabilities of models pretrained exclusively on image-only data (right) and those pretrained on image-text pairs (middle). The sole distinction between these models lies in their pretraining data type; all other hyperparameters and fine-tuning settings remain consistent. Despite not being fine-tuned on specific editing tasks, the model pretrained with image-only data demonstrates superior consistency with the original input image. For instance, in the first row, the right image closely resembles the raw input, while in the second and third rows, the right images maintain nearly identical gestures to the original.

## C.5 UMM FINETUNE RESULT

Tab. 10 show the detailed WISE score of the UMM finetuning results shown in Tab. 2.

Table 10: **UMM finetuning WISE results.** Notation A⊕B denotes the result obtained by combining methods A and B. ↓/↑ indicate a decrease/increase, respectively, in the metric compared to the baseline performance of the pretrained models.

| METHOD | Res. | NFEs | Cultural | Time | Space | Biology | Physics | Chemistry | Overall |
|---|---|---|---|---|---|---|---|---|---|
| OpenUni-L (Wu et al., 2025b) | 512 | 20×2 | 0.51 | 0.45 | 0.58 | 0.39 | 0.50 | 0.30 | 0.52 |
| ⊕Image finetuning | 512 | 20×2 | 0.46 | 0.52 | 0.66 | 0.49 | 0.51 | 0.29 | 0.49 |
| ⊕Pair finetuning | 512 | 20×2 | 0.63 | 0.58 | 0.74 | 0.57 | 0.71 | 0.44 | 0.62 |
| ⊕Mix finetuning | 512 | 20×2 | 0.60 | 0.58 | 0.70 | 0.51 | 0.64 | 0.46 | 0.59 |
| Qwen-Image (Wu et al., 2025a) | 512 | 50×2 | - | - | - | - | - | - | - |
| ⊕Image finetuning | 512 | 50×2 | 0.39 | 0.42 | 0.56 | 0.32 | 0.50 | 0.28 | 0.41 |
| ⊕Pair finetuning | 512 | 50×2 | 0.62 | 0.62 | 0.76 | 0.56 | 0.74 | 0.36 | 0.62 |
| ⊕Mix finetuning | 512 | 50×2 | 0.62 | 0.64 | 0.81 | 0.56 | 0.70 | 0.36 | 0.63 |
| Qwen-Image (Wu et al., 2025a) | 1024 | 50×2 | 0.62 | 0.63 | 0.77 | 0.57 | 0.75 | 0.40 | 0.62 |
| ⊕Image finetuning | 1024 | 50×2 | 0.28 | 0.35 | 0.52 | 0.40 | 0.40 | 0.28 | 0.35 |
| ⊕Pair finetuning | 1024 | 50×2 | 0.63 | 0.63 | 0.77 | 0.62 | 0.72 | 0.37 | 0.63 |
| ⊕Mix finetuning | 1024 | 50×2 | 0.64 | 0.63 | 0.78 | 0.57 | 0.73 | 0.38 | 0.63 |

## C.6 GENERATION RESULTS COMPARISON OF UMM FINETUNING

As illustrated in Fig. 7, fine-tuning enhances the model's performance on tasks requiring reasoning. Although the understanding module was frozen during fine-tuning, the model's improved alignment between images and text enables more accurate generation of desired details.

Conversely, fine-tuning with image-only data leads to degradation across nearly all scenarios. Fig. 8 compares outputs from models fine-tuned with image-only data (upper) and mixed data (lower). The image generated by the image-only fine-tuned model exhibits misalignment with the prompt;

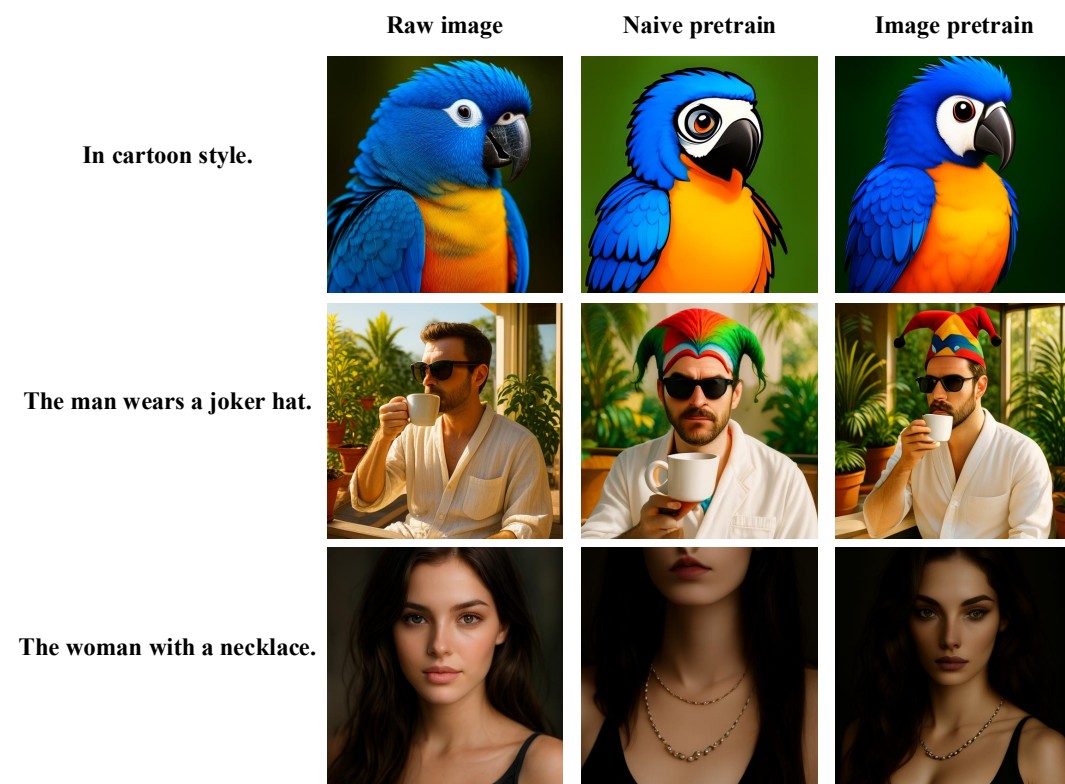

Figure 6: **Image editing ability with different pretraining method.**

for instance, in the first row, it depicts a black cat instead of a black sandwich. This misalignment underscores the importance of incorporating text-image pairs during fine-tuning to maintain semantic coherence.

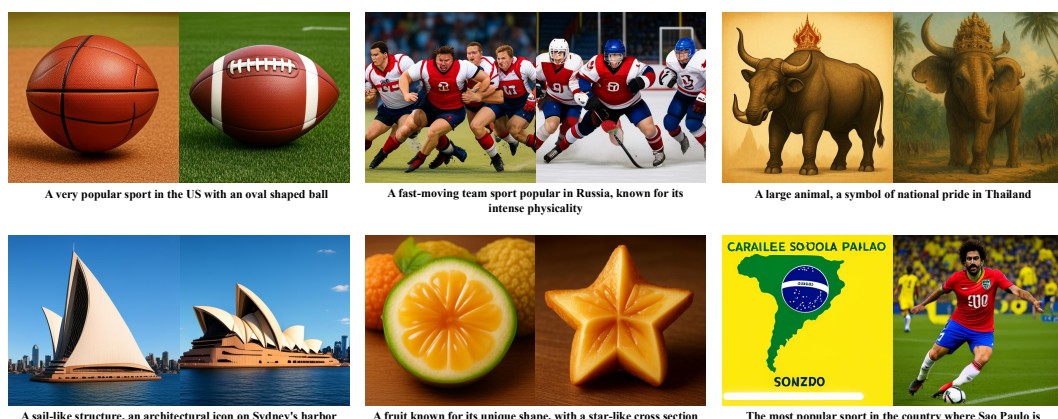

Figure 7: **Generation results of OpenUni-L before and after finetuning.** The left one is the image generated by the original OpenUni-L, while the right one is generated by the OpenUni-L after finetuning.

## C.7 PROMPTS DETAILS

The prompts used in Fig. 3 are as follows, from left to right, top to bottom.

- A british shorthair wearing sunglasses.

- A collection of vibrant red roses is artfully arranged on a rustic wooden surface. The roses, in full bloom, display their intricate petal layers and deep red hue, while the wooden background, with its visible grain patterns and knots, adds a textured contrast. The roses are placed in a cluster, with some overlapping others, creating a sense of depth and dimension.
- A fox wearing a suit and tie reading a newspaper at a café.
- A giant humanoid, made of fluffy blue cotton candy, stomping on the ground, and roaring to the sky, clear blue sky behind them.
- A glowing crystal ball floating above a sandstone table in the middle of a desert at sunset.
- A hot air balloon in the shape of a heart. Grand Canyon.
- A lighthouse standing alone in a stormy sea.
- A photo of a bear made entirely of autumn leaves.
- A renaissance-style oil portrait of a female astronaut wearing a richly ornate baroque spacesuit; deep chiaroscuro background with Rembrandt lighting, painterly brushwork but 32-bit color depth, captured in 16-K for museum-grade detail.
- A sea turtle swimming above a coral reef.
- A small hamster with a fluffy, light brown coat sits centrally on a red and orange striped sofa. The hamster's eyes are wide and alert, facing directly at the camera. The sofa's fabric features vertical stripes with a dark green and white border, creating a vibrant contrast with the hamster's soft fur. In the background, a dark green knitted blanket partially covers the sofa, adding texture to the scene.
- A stunning and luxurious bedroom carved into a rocky mountainside seamlessly blending nature with modern design with a plush earth-toned bed textured stone walls circular fireplace massive uniquely shaped window framing snow-capped mountains dense forests.
- A sunken ship at the bottom of the ocean.
- A young woman with freckles wearing a straw hat, standing in a golden wheat field.
- Close-up of a bright blue parrot's feathers glittering in the light, showing its unique plumage and vibrant colors.
- Close-up portrait of a young woman with light skin and long brown hair, looking directly at the camera. Her face is illuminated by dramatic, slatted sunlight casting shadows across her features, creating a pattern of light and shadow. Her eyes are a striking green, and her lips are slightly parted, with a natural pink hue. The background is a soft, dark gradient, enhancing the focus on her face. The lighting is warm and golden.
- Portrait of a beautiful, curvaceous, Pirate princess goddess babe, red hair, intricate ornate costume; Caribbean background + outdoors + Ocean, painted by ArtGerm, Alphonse Mucha, Roberto Ferri, Ross Tran, Pixar, low angle shot, digital painting: cinematic rim lighting, Unreal Engine 5, 8K.
- The reflection of a snowy mountainpeak in a crystal-clear alpine lake, creating a perfect mirror image with a slight shimmering effect.

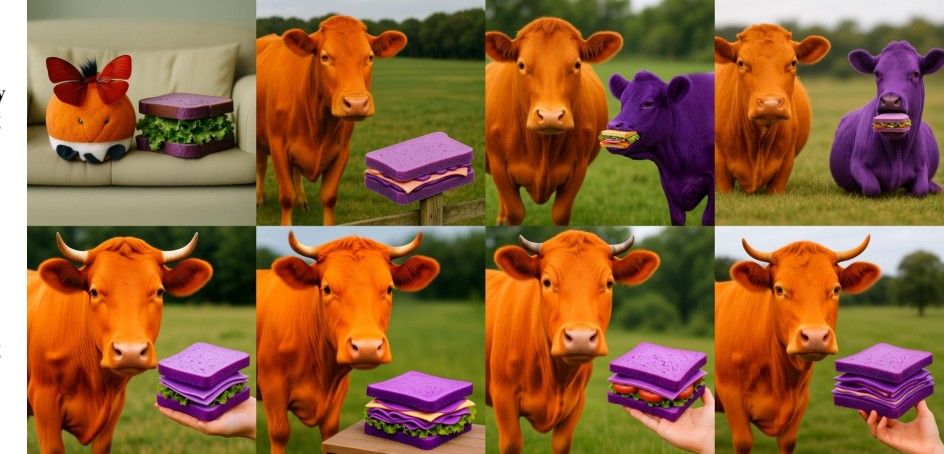

Figure 8: **Generation results comparison of Qwen-Image finetuned with image-only data and mixed data.**

