# OpenReview forum: "IOMM: Fast Pre-training of Unified Multimodal Models without Text-Image Pairs"
_ICLR.cc/2026/Conference — ICLR 2026 Conference Withdrawn Submission_

### Official Review · Reviewer_qaZE · 2025-10-27

**Soundness:** 3
**Presentation:** 3
**Contribution:** 3
**Rating:** 6
**Confidence:** 4

**Summary:**

This paper presents IOMM, a two-stage training paradigm for Unified Multimodal Models (UMMs) that enables efficient pre-training without reliance on large-scale text-image paired datasets. The approach involves an initial unsupervised image-only pre-training phase using a novel residual query adapter and masked image modeling, followed by fine-tuning that mixes image-only and text-image paired data. Extensive experiments on benchmark datasets demonstrate that IOMM achieves strong text-to-image generation and editing performance, sets new state-of-the-art or comparable results, scales favorably, and generalizes improvements to other open-source UMMs. Notably, the method provides significant efficiency (GPU hours) gains and shows emergent editing capability in a zero-shot setting.

**Strengths:**

1. Innovative Methodology: The two-stage approach—leveraging unsupervised image-only pre-training and a lightweight, parameter-efficient residual query adapter (RQA)—addresses practical challenges in pre-training large-scale UMMs without extensive paired data. The masked image modeling scheme is coherently integrated and well articulated (see Sec. 3 & Algorithm 1).

2. Experiment Results: Table 1 demonstrates strong results on GenEval (0.89 for IOMM-B) and competitive or superior outcomes on DPGBench and WISE, with thorough benchmarking against a suite of SOTA baselines, including some trained on much larger proprietary datasets.

3. Ablation Study: The efficacy of the RQA (see Fig. 5a), choice of mask ratio (Fig. 5b), and fine-tuning data mix (Fig. 5c) are explicitly quantified, supporting model design choices.

**Weaknesses:**

1. Ablation on Data Efficiency: While some comparisons are made to baseline models trained with more proprietary data, the reported experiments do not provide head-to-head runs with “paired-data-only” pre-training on the same dataset size and compute budget, limiting the interpretability of data efficiency claims (see Table 1/2 and Fig. 4).

2. Generalization is Limited: For the image-only setting, the authors only use the images from the same dataset as the text-image pair setting. If we already have the text-image pair data, the meaning of conducting image-only pretraining is limited. It would be better to conduct experiments on unsupervised large-scale natural image-only dataset to validate the effectiveness of such approach.

**Questions:**

1. An interesting observation is made in Figure 1 (c)-(d): when pre-training with paired data, performance is even worse than when paired or mixed fine-tuning is performed with image-only data pretraining. Conversely, when fine-tuning with images, paired pre-training significantly outperforms image-only pre-training. This result is somewhat counterintuitive and inconsistent. I hope the authors can provide more explanation for this and provide more experimental results on other benchmarks.

---

### Official Review · Reviewer_RTYs · 2025-10-31

**Soundness:** 2
**Presentation:** 2
**Contribution:** 2
**Rating:** 2
**Confidence:** 2

**Summary:**

This paper introduces IOMM, a two-stage training framework for Unified Multimodal Models (UMMs) that aims to reduce the dependency on large-scale, high-quality text-image paired data. The core idea is to first pre-train the model's generative component using only unlabeled images via a self-conditioning and masked image modeling objective. The second stage involves fine-tuning the model on a mixture of image-only data and a smaller set of paired data. The authors propose two technical components: a Residual Query Adapter (RQA) for efficient adaptation of a frozen MLLM, and a masked image modeling strategy to learn robust visual priors. Experiments show that their method achieves state-of-the-art (SOTA) performance on benchmarks like GenEval, while being data- and compute-efficient.

**Strengths:**

- The paper addresses a significant bottleneck in the development of UMMs: the heavy reliance on massive, often proprietary, text-image datasets. The goal of developing a data- and compute-efficient training paradigm that leverages abundant unlabeled image data is highly relevant and valuable to the community.
- A major strength of this work is the systematic investigation of six different training recipes (combinations of Image-only, Pair, and Mix data for pre-training and fine-tuning, as shown in Fig. 1c/d). This comparative analysis itself is a useful contribution, providing the community with valuable insights into the roles of different data compositions at different training stages.

**Weaknesses:**

While the paper presents a promising direction, there are several critical issues regarding novelty, experimental interpretation, and positioning within the existing literature that need to be addressed.
- Novelty and Positioning in Relation to Concurrent Work: The paper's primary contribution seems to have significant overlap with several very recent and highly relevant works, which are not cited or discussed. This raises serious concerns about the novelty of the proposed method.
  ○ Q1 (re: Lumos[1]): The core motivation and the finding that image-only pre-training benefits text-to-image generation are nearly identical to those in Lumos (a CVPR'25 paper). Lumos also leverages a vast amount of image-only data for pre-training to improve a UMM's generative abilities. How does IOMM's image-only self-conditioning approach fundamentally differ from the pre-training objectives used in Lumos, and what are the distinct advantages that justify IOMM as a novel contribution?
  ○ Q2 (re: Reconstruction Alignment[2]): The proposed mixed-data fine-tuning strategy appears to be a rediscovery of the method presented in "Reconstruction Alignment Improves Unified Multimodal Models" (arXiv). That paper explicitly proposes mixing reconstruction tasks (image-to-image, text-to-text) with cross-modal tasks (text-to-image) to improve model performance. The authors must cite this work and clarify what, if any, is novel about their "Mix" strategy compared to this existing method.
  ○ Q3 (re: Unified Multimodal Model as Auto-Encoder[3] & Visual Lexicon[4]): The fundamental idea of training a unified model with an image auto-encoding objective is a central theme in recent works like "Unified Multimodal Model as Auto-Encoder" and "Visual Lexicon". These works also aim to learn rich visual features through reconstruction. Could the authors provide a detailed discussion in the Related Work section that compares IOMM's architecture (frozen MLLM + RQA) and masked modeling objective against these methods to clearly delineate IOMM's unique contributions? The lack of this discussion makes it difficult to assess the paper's novelty.

[1] Learning Visual Generative Priors without Text

[2] RECONSTRUCTION ALIGNMENT IMPROVES UNIFIEDMULTIMODAL MODELS

[3] Unified Multimodal Model as Auto-Encoder

[4] Visual Lexicon: Rich Image Features in Language Space

- The experimental results, while strong overall, do not unequivocally support the central hypothesis that image-only pre-training is a superior paradigm.
  ○ Q4: According to Figure 1, the best recipe (Image-pretrain + Mix-finetune) achieves a GenEval score of 0.89. However, a more conventional approach (Pair-pretrain + Mix-finetune) achieves 0.88. An improvement of 0.01 is marginal and could potentially be within the range of experimental variance. Does this minor gain truly justify the strong claim of superiority for image-only pre-training, especially when the Image-pretrain + Image-finetune recipe performs extremely poorly (0.61)?
  ○ Q5: The results suggest that the mixed-data fine-tuning is the most crucial element for high performance, as it consistently provides the best results for both pre-training paradigms. Since this mixing strategy is not novel (be simialr to some existing methods), it appears that the paper's main practical contribution is the application of a known fine-tuning technique. How can the authors better disentangle the effects of the pre-training stage from the fine-tuning stage to more convincingly demonstrate the unique value added by the image-only pre-training itself? For example, what happens if the Pair-pretrain model is fine-tuned for more steps? Can it close the 0.01 gap?

- Lack of Evaluation on Multimodal Understanding:
  ○ Q6: The paper's central claim is about improving Unified Multimodal Models, which implies a synergy between understanding and generation. However, the entire experimental evaluation is focused on generative tasks (T2I generation, image editing). There is a critical lack of evaluation on the model's understanding capabilities. This makes the "unified" claim unsubstantiated. Have the authors evaluated the impact of their training paradigm on the underlying MLLM's understanding performance? For instance, presenting results on standard multimodal understanding benchmarks such as MME, MMBench, SEED-Bench, or even classic VQA benchmarks would be essential. It is crucial to demonstrate that the generative training does not degrade the model's understanding abilities (catastrophic forgetting). The authors state that freezing the MLLM prevents this (Sec 3.3); providing benchmark scores would be the direct and necessary proof for this claim.

**Questions:**

Please see my comments in weaknesses.

---

### Official Review · Reviewer_erqp · 2025-11-01

**Soundness:** 3
**Presentation:** 3
**Contribution:** 3
**Rating:** 6
**Confidence:** 4

**Summary:**

This paper proposes IOMM, a two-stage training paradigm for extending vision-language models (VLMs) to image generation tasks. It introduces a Residual Query Adapter (RQA) and adopts a masked image modeling strategy during image-only pretraining to further enhance the representation learning. Experimental results demonstrate that this training paradigm achieves strong performance across multiple benchmarks (e.g., GenEval, DPG-Bench, WISE) compared to other pretraining–fine-tuning recipes. Moreover, it exhibits good generalization when applied to existing unified multimodal generation models.

**Strengths:**

- The paper introduces a clear and practical two-stage training paradigm—image-only pretraining followed by mixed-data fine-tuning—that significantly reduces reliance on paired data.
- The proposed Residual Query Adapter (RQA) offers a lightweight, parameter-efficient way to adapt a frozen MLLM for generation, yielding substantial performance gains.
- Experiments show the proposed methods can achieves strong performance on various benchmarks.

**Weaknesses:**

- The improvement of RQA over MetaQuery may partly stem from increased parameters or FLOPs rather than genuine architectural advantages, as no capacity-matched comparison is provided.
- The work provides limited theoretical or mechanistic insight into why image-only pretraining plus mixed fine-tuning improves generation quality.
- The paper does not provide sufficient ablation studies, especially on the pretraining recipe, e.g. how different strategies and training steps influence the final performance.

**Questions:**

1. Can the authors offer any empirical or conceptual explanation for why image-only pretraining followed by mixed-data fine-tuning improves generative performance?
2. The authors should report the parameter count or flops of RQA and MetaQuery to rule out improvements caused by extra capacity. If they differ, please include a controlled comparison with matched params number or flops.
3. The related work describes a type of methods “integrating powerful MLLMs with established diffusion backbones.” However, Transfusion jointly trains LM and diffusion objectives from sratch in a unfied single transformer, rather than leveraging separate components. Could the authors clarify this classification?
4. The paper states that Fig. 4 compares models pre-trained on image-only and text-image data, but the figure only shows fine-tuning results—could the authors clarify this inconsistency?
5. The caption of Figure 4 states that panels (a–c) compare three data compositions, but the plots only show two (“Image” and “Pair”).

---

### Official Review · Reviewer_HA4f · 2025-11-02

**Soundness:** 2
**Presentation:** 2
**Contribution:** 2
**Rating:** 2
**Confidence:** 4

**Summary:**

This paper introduces IOMM, a novel and highly efficient two-stage training framework for Unified Multimodal Models (UMMs) that significantly reduces the conventional reliance on large-scale, text-image paired datasets. The core problem addressed is that training powerful UMMs is often bottlenecked by the need for vast, expensive, and often proprietary paired data, as well as immense computational resources. The proposed IOMM framework consists of an image-only pre-training stage and a mixed-data fine-tuning stage.  Experiments show that IOMM achieves SOTA performance on the GenEval benchmark, surpassing strong baselines.

**Strengths:**

The paper presents compelling results. The IOMM-B model achieves a state-of-the-art score of 0.89 on GenEval, outperforming well-established models like BAGEL.

**Weaknesses:**

-  Relying almost exclusively on GenEval and ImgEdit is insufficient to demonstrate the model's overall generative quality and generalization capabilities. I recommend including results from more challenging and diverse benchmarks, i.e., GenAI bench, OneIG bench, etc.
- The primary benchmark, GenEval, has a strong distributional overlap with the BLIP3-0-60K fine-tuning dataset. This calls into question whether the reported high scores are a true measure of performance or an artifact of the similar data distributions

**Questions:**

please see weaknesses

---

### Author Response · Authors · 2025-11-14

We would like to express our sincere gratitude to the reviewers and the program committee for their time and thoughtful feedback. We will carefully consider all their comments and revise our manuscript accordingly.

However, after deep reflection, we find ourselves unable to accept the evaluation we received. Despite the large amount of effort we have dedicated, we strongly believe in the significance of our **masking-modeling approach for pre-training unified multimodal models’ (UMMs) visual generation, especially under our image-only pretraining paradigm**, and yet we received two “2” scores. In light of this, we respectfully request to withdraw our submission.

Thank you again for the opportunity and for your understanding. We plan to refine our work and resubmit an improved version in the future.

---

### Note · Authors · 2025-11-14

I have read and agree with the venue's withdrawal policy on behalf of myself and my co-authors.